# Maternal vitamin D intake and BMI during pregnancy in relation to child's growth and weight status from birth to 8 years: a large national cohort study

Anna Amberntsson [1], Eleni Papadopoulou,[2] Anna Winkvist,[1] Lauren Lissner,[1] Helle Margrete Meltzer,[2] Anne Lise Brantsaeter [2] Hanna Augustin[1]

¹Institute of Medicine, University of Gothenburg, Goteborg, Sweden
²Norwegian Institute of Public Health, Oslo, Norway

**Correspondence to**
Anna Amberntsson;
anna.amberntsson@gu.se

## ABSTRACT

**Objectives** To examine the associations between maternal vitamin D intake and childhood growth and risk of overweight up to 8 years. We further examined the effect modification by maternal prepregnancy body mass index (BMI).

**Design** Prospective population-based pregnancy cohort study.

**Setting** The Norwegian Mother, Father and Child Cohort Study.

**Participants** In total, 58 724 mothers and 66 840 singleton children, with information on maternal vitamin D intake during the pregnancy and minimum one postnatal anthropometric measurement.

**Outcome measures** Predicted weight and height growth trajectories and velocities from 1 month to 8 years, rapid growth during infancy and toddlerhood, and risk of overweight in preschool and school age.

**Results** Overall, maternal vitamin D intake was associated with lower weight trajectory, lower odds of rapid weight growth and higher odds of childhood overweight. In children of mothers with prepregnancy normal weight, maternal vitamin D intake was negatively associated with weight trajectory and lower OR of a rapid weight growth during the first year, compared with reference (<5 µg/day). Children of mothers with normal weight, with maternal vitamin D intakes of 10–15 and >15 µg/day, also had 0.86 (95% CI 0.77 to 0.97) and 0.88 (95% CI 0.79 to 0.99) lower odds for overweight at 3 years, compared with reference. In contrast, in children of mothers with prepregnancy overweight (BMI ≥25 kg/m²), vitamin D intake was positively associated with weight trajectory. Children of mothers with overweight, with maternal vitamin D intake of 5–9.9 µg/day, also had (1.09 (95% CI 1.01 to 1.18) and 1.12 (95% CI 1.02 to 1.23)) higher odds for overweight at 5 years and 8 years, compared with reference.

**Conclusions** Maternal vitamin D intake affects postnatal growth and is inversely associated with childhood overweight in children of mothers with normal weight. Associations between maternal vitamin D intake and child growth and risk of overweight varied by prepregnancy BMI.

## STRENGTHS AND LIMITATIONS OF THIS STUDY

⇒ To our knowledge, this is the first study to investigate the association between maternal vitamin D intake during the pregnancy and postnatal growth.
⇒ The large, nationwide study population is a major strength.
⇒ Maternal vitamin D intake was estimated from both foods and supplements by a validated Food Frequency Questionnaire.
⇒ A limitation is self-reported maternal data and parental-reported growth data of the child

## INTRODUCTION

Vitamin D has received much attention over the last few decades and its role in human physiology seems to extend beyond bone health.[1] The human skin can synthesise a precursor of the vitamin when exposed to ultra violet-B rays of wavelength 290–315 nm.[2] At northern latitudes, this only occurs from April to October. Therefore, intake of vitamin D from foods and supplements becomes important during winter months in many countries, including the Nordics.[3] Vitamin D occurs naturally in a few foods, such as fatty fish, but is presently added to many fortified foods in the form of vitamin $D_3$.[2] The more uncommon form vitamin $D_2$ is only present in some plants and mushrooms.[4] Synthesised or ingested vitamin D (vitamin $D_3$ and vitamin $D_2$) is hydroxylated in the liver to 25-hydroxyvitamin D (25OHD), an established blood biomarker of vitamin D status. Further, 25OHD undergoes another hydroxylation primarily in the kidneys to form the physiologically active 1,25-dihydroxyvitamin D (1,25OH$_2$D).[4] The circulating concentration of 25OHD has shown to be inversely associated with body mass index (BMI), suggesting that individuals with obesity have a decreased bioavailability of vitamin D.[5]

Worldwide, the prevalence of vitamin D insufficiency (here defined as 25OHD<50 nmol/L) is high among pregnant women.[6] In some regions, such as South-East Asia and Western Pacific, the prevalence of vitamin D insufficiency were as high as 80%.[6] In contrast, in Europe and Eastern Mediterranean, the prevalence of vitamin D insufficiency were 57% and 46%, respectively.[6] In the Nordic countries, up to 28% of the pregnant fair-skinned European women have vitamin D insufficiency, but among non-fair skinned European women living in the Nordic countries, the prevalence is rising between 69% and 82%.[7–9] Therefore, on a population level, vitamin D insufficiency and deficiency in pregnancy are a relevant public health issue affecting the future generations. Pregnancy is a critical period where fetal exposure to poor conditions can affect later outcomes in the offspring.[10 11] Maternal vitamin D status, assessed as serum or plasma 25OHD concentrations, has been suggested as a potential modifiable early-life risk factor for development of childhood obesity.[12] Early rapid growth can increase the risk for childhood obesity, and both conditions are associated with several comorbidities in both childhood and adulthood, and is likely to follow the child through life.[13–16] In 2018, globally the prevalence of children with overweight and obesity under 5 years of age had risen to 40 million.[17]

Vitamin D contributes to the embryo implantation and is also important for placental function.[18] The maternal 25OHD easily passes the placenta and is further hydroxylated into $1,25OH_2D$ by the fetal kidneys, providing the fetus with vitamin D.[19] Vitamin D deficiency during pregnancy increases risk of developing rickets in the child.[20] Maternal vitamin D deficiency has also been associated with several other adverse neonatal outcomes, such as small for gestational age.[21] Associations have been found between low maternal vitamin D status and an increased risk of the child being born small for gestational age and with a lower birth weight,[22] as well as accelerated growth in weight and height during infancy.[23] Maternal vitamin D status has also been inversely associated with childhood adiposity at 4 and 6 years[24] and childhood insulin resistance at 9.5 years.[25] Further, associations of an increased risk of childhood overweight at 12 months of age was seen in children of mothers with low vitamin D status during the pregnancy.[26 27] However, some studies have found opposite results, that is, that poor maternal vitamin D status was associated with reduced child growth,[28] while others have reported null associations with childhood adiposity[25 29 30] and growth.[26 29] Even though the vitamin D intake from food and supplements is a determinant of vitamin D status, the role of maternal vitamin D intake during pregnancy has been less studied and the limited evidence regarding fetal growth is discordant.[31 32] Associations between maternal vitamin D intake in pregnancy and postnatal growth and overweight have not yet been explored. Since vitamin D intake affects vitamin D status,[4] and due to the decreased bioavailability of vitamin D in obesity, it is possible that such an association could be modified by maternal prepregnancy BMI.

Our objectives were to assess the associations between maternal vitamin D intake during the pregnancy and childhood weight and height growth trajectories and velocities, rapid growth and risk of overweight up to 8 years, in a large, prospective pregnancy cohort study. Further, we aimed to examine maternal prepregnancy BMI as an effect modifier.

## METHOD
### Study population
Our study draws resources from the Norwegian Mother, Father and Child Cohort Study (MoBa), a population-based pregnancy cohort study conducted by the Norwegian Institute of Public Health (NIPH).[33] Participants were recruited from all over Norway during 1999–2008 and invited at the time of their first ultrasound scanning. The women consented to participation in 41% of the pregnancies. The study now includes 95 200 mothers, 75 200 fathers and 114 500 children. The current study is based on version 12 of the quality-assured data files released for research in January 2019. The establishment of MoBa and initial data collection was based on a license from the Norwegian Data Protection Agency and approval from The Regional Committees for Medical and Health Research Ethics. The MoBa cohort is now based on regulations related to the Norwegian Health Registry Act.

Questionnaires were sent out to the participating women in pregnancy week 15, week 22 and week 30. The questionnaires in week 15 and 30 requested background information, such as education and parity, whereas the questionnaire in week 22 was a Food Frequency Questionnaire (FFQ). After delivery, follow-up of the participants by questionnaires was conducted at six different time points; 6 months, 18 months, 36 months, 5 years, 7 years and 8 years.[33 34] Participants in MoBa received all questionnaires in Norwegian, and lack of language skills was the only exclusion criterion. However, the questionnaires are now available in English translation on the NIPH website.[35] The MoBa database is linked to the Medical Birth Registry of Norway, which is a national health registry containing information about all births in Norway.[36]

Exclusion criteria for this study were stillbirth, multiple gestation, malformation and chromosomal abnormalities. Of the eligible 105 344 mother–child pairs, further exclusion was done due to no available data on vitamin D intake during pregnancy, child's gender and birth weight and length, and implausible maternal energy intakes (<1075 kcal and >4780 kcal per day).[37] The final study population included 66 840 mother-child pairs (58 724 mothers and 66 840 children) with available information on all covariates and at least one postnatal measurement on weight and length/height. There are more children than mothers because 22 039 of the children were siblings. The retention rate of our study population by follow-up is presented in online supplemental table 1.

## Patient and public involvement

Neither patients nor the public were involved in the design and conduct of this study. The cohort is ongoing and information about research projects and results from MoBa are available to participants, researchers and the public on the NIPH website.[38]

## Maternal vitamin D intake

Maternal diet during pregnancy was assessed by a validated semiquantitative FFQ, during gestational week 22. This FFQ was used in the MoBa study from the year 2002 and onwards.[37] Pregnant women were asked to report how often they consumed 225 different food and beverage items and their supplement use since becoming pregnant. The FFQ was tailored to capture dietary habits and supplement use during the first half of pregnancy. Of all the participating women in MoBa, 93% answered the FFQ.[37] A validation study in a subset of 119 MoBa participants demonstrated satisfactory validity for FFQ intakes and fair ability of the FFQ to rank individuals according to food, energy and nutrient intakes using a 4-day weighed food dairy and biological markers of food intake as reference methods.[39 40] The results showed that vitamin D supplement use was the dominant predictor of plasma 25OHD, whereas vitamin D from foods was not associated with 25OHD. The majority of participants were recruited in the months with low sun exposure (October to April), but even within this range seasonal variation in plasma 25OHD could be observed.[40] The Spearman's rank correlation coefficient between maternal vitamin D intake and plasma 25OHD concentration was r=0.32 (p<0.01) for all participants in the validation study (n=119) and r=0.45 (p<0.01) when only individuals recruited during October to April (n=81) were included.[40] Total vitamin D (from foods and supplements), vitamin D from foods alone and vitamin D from supplements alone were estimated from the FFQ and used in our study.

Maternal vitamin D intake was categorised according to national recommended intakes at the time of MoBa recruitment as well as according to current recommendations.[41] Total vitamin D intake (from foods and supplements), and vitamin D intake from supplements alone, were categorised into; <5 µg/day, 5–9.9 µg/day, 10–15 µg/day and >15 µg/day. The categories for vitamin D intake from foods alone were: <2.5 µg/day, 2.5–4.9 µg/day, 5.0–7.5 µg/day and >7.5 µg/day.

## Childhood anthropometry

Measurements of weight and height and the dates of the measurements of the children were obtained at eleven ages (6 weeks, 3 months, 6 months and 8 months and 1 year, 1.5 years, 2 years, 3 years, 5 years, 7 years and 8 years) by six questionnaires. During the first 18 months postpartum, the mothers were asked to report the child's documented anthropometric measures and dates from the health card, but no specification was made for measurements onwards. There were on average seven reported measurements of weight and height,

respectively, per child. From the available reported growth measurements, individual growth trajectories for weight and height were predicted using the Jenss-Bayley growth curve model[42 43] with the Saemix algorithm[43] applied. Hence, the predicted measures of weight and height were obtained on the dates of the reported anthropometrics. Implausible values of the reported anthropometrics were identified and removed if they deviated ≥3 SD from the predicted value, derived from the growth curve model. About 3% (n=50 331) of the weight and height values were excluded as implausible.

## Outcomes

After exclusion of implausible reported anthropometric values, individual growth trajectories of weight and height, as well as weight growth velocity (g/month) and height growth velocity (cm/month) were predicted at nine different age points (1 month, 3 months, 6 months, 1 year, 1.5 years, 2 years, 3 years, 5 years and 8 years). The trajectories and velocities were predicted at the exact age points (the exact day the child turned 1 month, 3 months, etc), separately for girls and boys, in order to compensate for the loss to follow-up of anthropometric measures. By not including birth weight and length in the models of growth, the effect of maternal vitamin D intake on growth could be assessed independently of its potential effect on birth size,[31 32] as birth size might influence the estimated growth trajectories.

Rapid weight and height growth in infancy and toddlerhood (up to 2 years) were evaluated using conditional growth modelling, an approach to obtain growth measures of weight and height independent of their prior measurements.[44] Reported anthropometrics were used for the calculation of conditional weight and height at 12 months and at 2 years. Standardised residuals were calculated by a regression of the anthropometric measure of interest, adjusting for its prior measurements. Rapid weight or height growth were defined as the residual deviating >1 SD from the expected value. All other children were defined as non-rapid growers and were used as the reference.

Children's BMI values were calculated at preschool age (3 years and 5 years) and school age (8 years) using the predicted values from the growth model. By using the extended International Obesity Task Force gender-specific cut offs,[45] overweight and obesity were defined. The outcome childhood overweight includes obesity (further denoted as overweight). The BMI cut-offs used for categorisation at the different ages are presented in online supplemental table 2.

## Statistical analysis

Differences in characteristics of the study population by categories of maternal vitamin D intake were investigated using $\chi^2$ test for categorical variables and analysis of variance for continuous variables. The association between maternal vitamin D intake and predicted weight and height growth trajectories and velocities at the nine time

points were examined by multilevel mixed effects linear regression model, with a random intercept by child and random slope for age. The covariates were included as fixed in the mixed effect models. Logistic regression was used to investigate the association between total maternal vitamin D intake and the risk to experience rapid growth in infancy and toddlerhood, and risk of overweight in preschool and school ages. The lowest categories of maternal vitamin D intake was used as the reference group (for total intake and for supplementary intake) <5 µg/day, and for food intake <2.5 µg/day. Our primary exposure was total maternal vitamin D intake and in secondary analyses, we examined intakes from food and supplements separately.

By drawing a directed acyclic graph, hypotheses about the causal paths were set and potential confounders identified (online supplemental figure 1). Based on the pathways in the graph, models were adjusted for maternal education as a proxy for socioeconomic position, parity, maternal milk and yoghurt intake, maternal fibre intake, maternal prepregnancy BMI and child's gender. Maternal age, maternal energy intake and birth weight were also considered but not included in the final models, as these variables did not change the estimated effect. Since some mothers participated in more than one mother–child pair, all regression models were also adjusted for random effects of sibling clusters. Potential interaction variables tested were maternal prepregnancy BMI (in kg/m²)[46 47] and birth weight.[32 48] Interaction were considered if p<0.100 for the included interaction variable. The only significant interaction variable was prepregnancy BMI (p<0.001). Therefore, results are presented for all mothers, and separately for children of mothers with prepregnancy BMI 18.5–24.9 kg/m² (further denoted as mothers with normal weight), and for children of mothers with prepregnancy BMI ≥25 kg/m² (further denoted as mothers with overweight). Maternal age was divided into four categories (<20, 20–29, 30–39, ≥40 years). Maternal education was divided into three categories (<13, 13–16 and >16 years). Parity was categorised as primiparous or multiparous, for women without and with children from previous than the index pregnancies, respectively. Maternal prepregnancy BMI was divided into four categories (<18.5, 18.5–24.9, 25–29.9 and ≥30 kg/m²), according to WHO.[49] Smoking during pregnancy was categorised as never and ever.

Three sensitivity analyses were performed: (1) using the reported measures of weight and length/height instead of the predicted anthropometrics, (2) after excluding mothers with total vitamin D intakes >90th percentile (18.8 µg/day), (3) adjustment for birth weight in the childhood overweight models.

As an approach to estimate causal effects, discordant sibling analysis was conducted, with the purpose to control for unmeasured familial confounding.[50 51] In total, there were 22 039 siblings identified in the study subpopulation. For all women who participated with more than one pregnancy, mean maternal total vitamin D intake by family was calculated, as well as each sibling's deviation from its family mean. A cut-off at 5 µg/day was used, as this corresponds to the range of the maternal vitamin D categories. Discordant siblings were defined as having a maternal vitamin D intake deviating more than, or less than the cut-off from its family mean. Finally, weight and height growth trajectories, and the risk of overweight at 3, 5 and 8 years, were compared between siblings categorised as discordant (n=2346). The sibling in the lower vitamin D category was used as a reference.

The main analyses were performed using Stata V.16 statistical software (Stata), and RStudio V.1.2.5042 was used for the growth models.

## RESULTS

### Maternal characteristics and vitamin D intake

For mothers taking a vitamin D supplement, the supplement contributed overall with 68% of the total vitamin D intake. Total maternal vitamin D intake was positively associated with maternal age, education, vitamin D supplement use in pregnancy, gestational weight gain and maternal energy intake and negatively associated with parity, prepregnancy BMI and maternal smoking in pregnancy (table 1). In 31% of the pregnancies, the mother had a BMI≥25 kg/m². The median (25th–75th centiles) total vitamin D intake from foods and supplements was 7.2 µg/day (4.2–12.1 µg/day) for the included participants and 6.7 µg/day (3.9–11.7 µg/day) for the excluded group (n=13 826, p<0.001) (data not shown). The median vitamin D intake from foods alone was 3.1 µg/day (2.0–4.4 µg/day) for the included participants and 3.1 µg/day (2–4.5 µg/day) for the excluded group, with available information on vitamin D intake (n=16 872 mothers, p=0.589).

### Total maternal vitamin D intake during pregnancy and childhood growth

#### Weight and height growth trajectories and velocities from 1 month to 8 years

Overall, total maternal vitamin D intake was negatively associated with child weight growth from 1 month to 8 years for intakes ≥10 µg/day, compared with low intakes (<5 µg/day, reference group) (tables 2 and 3). Prepregnancy BMI modified the association between the exposure and the outcome (p value for interaction <0.001). More specifically, among mothers with normal prepregnancy BMI, we found that higher maternal vitamin D intake was negatively associated with child's weight, especially during the first 3 months, compared with low intakes (tables 2 and 3). The negative association with child's weight persisted for intakes of 10–15 µg/day and for up to 2 years, while no further reduction in weight was observed for higher intakes and after 3 months (>15 µg/day). Similar negative associations with weight from 3 months to 2 years were found when we examined maternal vitamin D intake from supplements only (online supplemental tables 3 and 4). In addition, maternal vitamin D intakes of 10–15 µg/day

**Table 1** Study population characteristics by categories of total maternal vitamin D intake during pregnancy (n=66 840)

| | Vitamin D intake <5µg/day N=21 748 | | Vitamin D intake 5–9.9 µg/day N=22 718 | | Vitamin D intake 10–15 µg/day N=11 143 | | Vitamin D intake >15µg/day N=11 231 | | All N=66 840 | | P value* |
|---|---|---|---|---|---|---|---|---|---|---|---|
| | N | % | N | % | N | % | N | % | N | % | |
| **Maternal age (years)** | | | | | | | | | | | <0.001 |
| <20 | 194 | 0.9 | 135 | 0.6 | 83 | 0.7 | 82 | 0.7 | 494 | 0.7 | |
| 20–29 | 9566 | 44 | 9740 | 42.9 | 4684 | 42.0 | 4415 | 39.3 | 28 405 | 42.5 | |
| 30–39 | 11 622 | 53.4 | 12 449 | 54.9 | 6120 | 54.9 | 6429 | 57.3 | 36 620 | 54.8 | |
| ≥40 | 366 | 1.7 | 394 | 2.3 | 256 | 2.3 | 305 | 2.7 | 1321 | 2.0 | |
| **Maternal education (years)** | | | | | | | | | | | <0.001 |
| <13 | 7480 | 34.4 | 6371 | 28.1 | 2923 | 26.2 | 2822 | 25.1 | 19 596 | 29.3 | |
| 13–16 | 9238 | 42.5 | 10 138 | 44.6 | 4954 | 44.5 | 4899 | 43.6 | 29 229 | 43.7 | |
| >16 | 5030 | 23.1 | 6209 | 27.3 | 3266 | 29.3 | 3510 | 31.3 | 18 015 | 27.0 | |
| **Parity** | | | | | | | | | | | <0.001 |
| Primiparous | 8733 | 40.2 | 10 546 | 46.4 | 5469 | 49.1 | 5677 | 50.6 | 30 425 | 45.5 | |
| Multiparous | 13 015 | 59.8 | 12 172 | 53.6 | 5674 | 50.9 | 5554 | 49.4 | 36 415 | 54.5 | |
| **Prepregnancy BMI (kg/m²)** | | | | | | | | | | | <0.001 |
| <18.5 | 534 | 2.5 | 685 | 3 | 336 | 3 | 358 | 3.2 | 1913 | 2.9 | |
| 18.5–24.9 | 13 320 | 61.2 | 15 087 | 66.4 | 7599 | 68.2 | 7974 | 71 | 43 980 | 65.8 | |
| 25–29.9 | 5385 | 24.8 | 4870 | 21.4 | 2273 | 20.4 | 2152 | 19.2 | 14 680 | 22 | |
| ≥30 | 2509 | 11.5 | 2076 | 9.1 | 935 | 8.4 | 747 | 6.7 | 6267 | 9.4 | |
| **Maternal smoking in pregnancy** | | | | | | | | | | | <0.001 |
| Never | 17 759 | 90.7 | 19 278 | 92.7 | 9643 | 93.8 | 9707 | 93.8 | 56 387 | 92.4 | |
| Ever | 1826 | 9.3 | 1525 | 7.3 | 640 | 6.2 | 648 | 6.2 | 4628 | 7.6 | |
| **Maternal vitamin D supplement use** | 9134 | 42 | 20 446 | 90 | 11 031 | 99 | 11 175 | 99.5 | 51 667 | 77.3 | <0.001 |
| **Child's gender** | | | | | | | | | | | 0.198 |
| Girls | 10 711 | 49.3 | 11 122 | 49 | 5385 | 48.3 | 5412 | 48.2 | 32 630 | 48.8 | |
| Boys | 11 037 | 50.7 | 11 596 | 51 | 5758 | 51.7 | 5819 | 51.8 | 34 210 | 51.2 | |
| | Median | Range† | Median | Range† | Median | Range† | Median | Range† | Median | Range† | |
| **Total maternal vitamin D intake (µg/day)** | 3.3 | 1.1–4.8 | 7.1 | 5.2–9.7 | 12.0 | 10.2–14.6 | 19.8 | 15.4–33.6 | 7.2 | 1.8–23.0 | <0.001 |
| Gestational age (weeks) | 40.1 | 37.3–42.1 | 40.1 | 37.3–42.1 | 40.3 | 37.3–42.1 | 40.3 | 37.3–42.1 | 40.1 | 37.3–42.1 | 0.224 |
| Gestational weight gain (kg) | 14.2 | 5–25 | 14.5 | 6–25 | 14.4 | 6–25 | 15.0 | 6–25 | 14.5 | 6–25 | <0.001 |
| Birth weight (g) | 3630 | 2780–4480 | 3620 | 2780–4470 | 3620 | 2800–4430 | 3620 | 2780–4460 | 3620 | 2780–4464 | <0.001 |
| Maternal energy intake (kcal) | 2092 | 1384–3166 | 2261 | 1491–3462 | 2321 | 1521–3558 | 2324 | 1527–3631 | 2222 | 1457–3436 | <0.001 |

*$\chi^2$ test for categorical variables and ANOVA for continuous variables.
†5–95th percentile.
ANOVA, analysis of variance.

**Table 2** Associations between total maternal vitamin D intake and childhood weight and height trajectories from 1 month to 2 years, using predicted anthropometrics

| Maternal vitamin D intake | Infancy | | | Toddlerhood | | |
|---|---|---|---|---|---|---|
| | 1 month Beta (95% CI) | 3 months Beta (95% CI) | 6 months Beta (95% CI) | 12 months Beta (95% CI) | 18 months Beta (95% CI) | 2 years Beta (95% CI) |
| **All mothers n=66 840** | | | | | | |
| Weight (g) | | | | | | |
| <5 µg (ref) | | | | | | |
| 5–9.9 µg | −9.4 (−19.1 to 0.2) | −9.2 (−19.9 to 0.6) | −8.8 (−19.2 to 1.5) | −8.1 (−20.7 to 4.4) | −7.4 (−23.1 to 8.3) | −6.7 (−26.0 to 12.5) |
| 10–15 µg | −7.6 (−19.5 to 4.3) | −9.3 (−21.3 to 2.8) | −11.7 (−24.5 to 1.0) | −16.7 (−32.2 to −1.2) | −21.6 (−40.9 to −2.3) | −26.6 (−50.3 to −2.9) |
| >15 µg | −14.7 (−26.7 to −2.7) | −15.4 (−27.6 to −3.3) | −16.6 (−29.5 to −3.8) | −19.0 (−34.5 to −3.5) | −21.4 (−40.7 to −2.1) | −23.7 (−47.4 to 0.1) |
| Height (cm) | | | | | | |
| <5 µg (ref) | | | | | | |
| 5–9.9 µg | −0.01 (−0.07 to 0.04) | −0.01 (−0.06 to 0.04) | −0.01 (−0.06 to 0.04) | −0.01 (−0.06 to 0.04) | −0.01 (−0.05 to 0.04) | −0.001 (−0.05 to 0.04) |
| 10–15 µg | −0.05 (−0.11 to 0.01) | −0.05 (−0.11 to 0.01) | −0.05 (−0.11 to 0.01) | −0.05 (−0.11 to 0.003) | −0.06 (−0.11 to −0.002) | −0.06 (−0.11 to −0.01) |
| >15 µg | −0.04 (−0.10 to 0.03) | −0.04 (−0.10 to 0.03) | −0.03 (−0.09 to 0.03) | −0.03 (−0.09 to 0.03) | −0.02 (−0.08 to 0.03) | −0.02 (−0.07 to 0.04) |
| **Mothers with normal weight* n=41 970** | | | | | | |
| Weight (g) | | | | | | |
| <5 µg (ref) | | | | | | |
| 5–9.9 µg | −11.5 (−22.4 to −0.6) | −10.9 (−22.0 to 0.14) | −10.1 (−21.8 to 1.6) | −8.4 (−22.5 to 5.8) | −6.7 (−24.3 to 10.9) | −5.0 (−26.6 to 16.6) |
| 10–15 µg | −14.5 (−27.7 to −1.2) | −15.7 (−29.1 to −2.3) | −17.5 (−31.7 to −3.3) | −21.1 (−38.2 to −4.0) | −24.7 (−46.0 to −3.5) | −28.4 (−54.5 to −2.3) |
| >15 µg | −15.2 (−28.3 to −2.1) | −14.5 (−27.8 to −1.2) | −13.5 (−27.5 to 0.6) | −11.4 (−28.3 to 5.5) | −9.3 (−30.4 to 11.7) | −7.3 (−33.1 to 18.5) |
| Height (cm) | | | | | | |
| <5 µg (ref) | | | | | | |
| 5–9.9 µg | −0.02 (−0.08 to 0.04) | −0.02 (−0.08 to 0.04) | −0.02 (−0.07 to 0.04) | −0.02 (−0.07 to 0.04) | −0.01 (−0.06 to 0.04) | −0.01 (−0.06 to 0.04) |
| 10–15 µg | −0.07 (−0.14 to −0.003) | −0.07 (−0.14 to −0.005) | −0.08 (−0.14 to −0.01) | −0.08 (−0.15 to −0.02) | −0.09 (−0.15 to −0.03) | −0.09 (−0.15 to −0.03) |
| >15 µg | −0.05 (−0.12 to 0.02) | −0.05 (−0.12 to 0.02) | −0.05 (−0.11 to 0.02) | −0.04 (−0.11 to 0.02) | −0.04 (−0.09 to 0.02) | −0.03 (−0.09 to 0.03) |
| **Mothers with overweight† n=20 080** | | | | | | |
| Weight (g) | | | | | | |
| <5 µg (ref) | | | | | | |
| 5–9.9 µg | −5.5 (−19.8 to 8.8) | −1.4 (−15.9 to 13.1) | 4.7 (−10.6 to 20.0) | 16.9 (−1.6 to 35.4) | 29.2 (6.1 to 52.2) | 41.4 (13.1 to 69.7) |
| 10–15 µg | 10.3 (−8.0 to 28.6) | 14.4 (−4.1 to 33.0) | 20.6 (1.0 to 40.2) | 33.1 (9.3 to 56.8) | 45.5 (15.9 to 75.0) | 57.9 (21.7 to 94.1) |
| >15 µg | −15.6 (−34.8 to 3.7) | −9.9 (−29.5 to 9.6) | −1.5 (−22.1 to 19.1) | 15.4 (−9.5 to 40.3) | 32.3 (1.2 to 63.3) | 49.2 (11.1 to 87.3) |
| Height (cm) | | | | | | |
| <5 µg (ref) | | | | | | |
| 5–9.9 µg | −0.02 (−0.09 to 0.06) | −0.02 (−0.09 to 0.06) | −0.01 (−0.08 to 0.06) | −0.005 (−0.07 to 0.06) | 0.002 (−0.06 to 0.06) | 0.009 (−0.06 to 0.07) |

Continued

**Table 2** Continued

| Maternal vitamin D intake | Infancy | | | Toddlerhood | | |
|---|---|---|---|---|---|---|
| | 1 month | 3 months | 6 months | 12 months | 18 months | 2 years |
| | Beta (95% CI) | Beta (95% CI) | Beta (95% CI) | Beta (95% CI) | Beta (95% CI) | Beta (95% CI) |
| 10–15µg | −0.02 (−0.11 to 0.08) | −0.02 (−0.11 to 0.08) | −0.01 (−0.10 to 0.09) | −0.003 (−0.09 to 0.09) | 0.005 (−0.08 to 0.09) | 0.01 (−0.07 to 0.10) |
| >15µg | −0.04 (−0.14 to 0.07) | −0.03 (−0.13 to 0.07) | −0.03 (−0.12 to 0.07) | −0.01 (−0.11 to 0.08) | −0.001 (−0.09 to 0.09) | 0.01 (−0.08 to 0.10) |

Effect estimates derive from multilevel mixed effects linear regression model, adjusted for maternal education, parity, maternal milk and yoghurt intake, maternal fibre intake, birth weight and child's gender. The model with all mothers was also adjusted for maternal prepregnancy BMI.

*Mothers with prepregnancy BMI 18.5–24.9 kg/m².

†Mothers with prepregnancy BMI ≥25 kg/m².

BMI, body mass index.

were associated with a lower height trajectory, compared with lower intakes, while no other intake level was associated with the child's height. No association was found between maternal vitamin D intake and child's weight or height growth velocity from 1 month to 8 years of age, in children of mothers with normal weight (online supplemental tables 8 and 9).

Among children of mothers with overweight prepregnancy BMI, any maternal vitamin D intake was positively associated with weight trajectory from 12 months onwards, compared with lower intakes (tables 2 and 3). Nevertheless, for intakes between 10–15 µg/day, the positive association with child's weight growth was already observed from 6 months and onwards. Similar positive associations with child weight after 3 years were found when we examined the intake from food and supplements separately, but for different intake levels (online supplemental tables 3 and 4). For height growth trajectory, the only significant difference from the reference group was a positive association in children prenatally exposed to vitamin D intake of >15 µg/day at 7 and 8 years of age, but the effect estimate was small. No association was found between maternal vitamin D intake and weight or height growth velocity from 1 month to 8 years of age, in children of mothers with overweight (online supplemental tables 8 and 9).

The sensitivity analyses on the association between maternal vitamin D intake and reported anthropometrics, gave similar results as the main models, in both children of mothers with normal weight and children of mothers with overweight (results not shown). Also, when excluding the highest 90th percentile of total maternal vitamin D intake, the associations seen in the main models on child's weights and heights remained.

### Risk of rapid weight and height growth in infancy and toddlerhood

Overall, maternal vitamin D intakes of 5–9.9 and >15 µg/day was associated with lower odds of having a rapid weight gain at 12 months of age, both crude and adjusted, compared with low intakes (<5 µg/day) (table 4).

When exploring the association by maternal prepregnancy BMI, children of mothers with normal prepregnancy weight, with total maternal vitamin D intakes of 5–9.9, 10–15 and >15 µg/day during pregnancy, had lower odds of having a rapid weight gain during the first year of life, both crude and adjusted compared with the reference group (table 4). Similar lower OR were found for the risk of rapid height gain from birth to 12 months, but the CI included 1. The associations with rapid weight and height gain from birth to 2 years were weaker.

For children of mothers with prepregnancy overweight, we found higher OR for rapid weight and height gain at 12 months and 2 years, both crude and adjusted, but these associations were weak and the confidence intervals included 1 (table 4).

**Table 3** Associations between total maternal vitamin D intake and childhood weight and height trajectories from 3 years to 8 years, using predicted anthropometrics

| | Preschool age | | School age |
|---|---|---|---|
| | 3 years | 5 years | 8 years |
| Maternal vitamin D intake | Beta (95% CI) | Beta (95% CI) | Beta (95% CI) |
| **All mothers n=66 840** | | | |
| Weight (g) | | | |
| *<5*µg (ref) | | | |
| 5–9.9µg | −5.3 (−32.4 to 21.8) | −2.5 (−46.1 to 41.2) | 1.8 (−67.3 to 70.9) |
| 10–15µg | −36.5 (−69.7 to −3.2) | −56.3 (−110 to −2.7) | −86.0 (−171 to −1.1) |
| >15µg | −28.5 (−61.7 to 4.8) | −37.9 (−91.4 to 15.5) | −52.1 (−137 to 32.5) |
| Height (cm) | | | |
| <5µg (ref) | | | |
| 5–9.9µg | 0.01 (−0.04 to 0.05) | 0.02 (−0.03 to 0.07) | 0.04 (−0.03 to 0.11) |
| 10–15µg | −0.06 (−0.11 to −0.01) | −0.07 (−0.13 to −0.01) | −0.09 (−0.17 to 0.003) |
| >15µg | −0.01 (−0.06 to 0.05) | 0.02 (−0.04 to 0.08) | 0.05 (−0.04 to 0.14) |
| **Mothers with normal weight* n=41 970** | | | |
| Weight (g) | | | |
| <5µg (ref) | | | |
| 5–9.9µg | −1.6 (−31.9 to 28.7) | 5.2 (−43.6 to 54.0) | 15.3 (−62.0 to 92.6) |
| 10–15µg | −35.6 (−72.3 to 1.0) | −50.1 (−109 to 8.9) | −71.8 (−165 to 21.6) |
| >15µg | −3.1 (−39.3 to 33.0) | 5.1 (−53.0 to 63.3) | 17.6 (−74.6 to 110) |
| Height (cm) | | | |
| <5µg (ref) | | | |
| 5–9.9µg | −0.004 (−0.05 to 0.05) | 0.008 (−0.05 to 0.06) | 0.03 (−0.06 to 0.11) |
| 10–15µg | −0.1 (−0.16 to −0.04) | −0.1 (−0.19 to −0.05) | −0.1 (−0.25 to −0.05) |
| >15µg | −0.02 (−0.08 to 0.04) | −0.003 (−0.07 to 0.07) | 0.03 (−0.07 to 0.12) |
| **Mothers with overweight† n=20 080** | | | |
| Weight (g) | | | |
| <5µg (ref) | | | |
| 5–9.9µg | 65.8 (26.1 to 106) | 115 (50.8 to 179) | 188 (86.8 to 290) |
| 10–15µg | 82.7 (31.9 to 134) | 132 (50.5 to 214) | 207 (77.2 to 337) |
| >15µg | 82.9 (29.5 to 136) | 151 (64.4 to 237) | 252 (116 to 388) |
| Height (cm) | | | |
| <5µg (ref) | | | |
| 5–9.9µg | 0.02 (−0.04 to 0.09) | 0.05 (−0.02 to 0.13) | 0.09 to (−0.01 to 0.20) |
| 10–15µg | 0.03 (−0.05 to 0.11) | 0.06 (−0.03 to 0.16) | 0.1 (−0.02 to 0.25) |
| >15µg | 0.04 (−0.05 to 0.12) | 0.09 (−0.01 to 0.19) | 0.2 (0.02 to 0.31) |

Effect estimates derive from multilevel mixed effects linear regression model, adjusted for maternal education, parity, maternal milk and yoghurt intake, maternal fibre intake, birth weight and child's gender. The model with all mothers was also adjusted for maternal prepregnancy BMI.
*Mothers with prepregnancy BMI 18.5–24.9 kg/m$^2$.
†Mothers with prepregnancy BMI ≥25 kg/m$^2$.
BMI, body mass index.

In the sensitivity analyses, when excluding the 90th percentile of maternal vitamin D intakes, similar results were obtained as in the main models, for both children of mothers with normal weight, and for children of mothers with overweight.

## Risk of childhood overweight in preschool and school age
Overall, the prevalence of childhood overweight was 8.9% at 3 years, 18.2% at 5 years and 6.3% at 8 years of age. Maternal vitamin D intakes of 5–9.9 and >15 µg/day were associated with a higher risk of overweight at 8 years of age, compared with the reference group (table 5). However, in children of mothers with normal weight, with total vitamin D intakes of 10–15 and >15 µg/day during pregnancy, 0.86 (95% CI 0.77 to 0.97) and 0.88 (95% CI 0.79 to 0.99) lower odds for overweight at 3 years of age were found, both crude

**Table 4** Associations (OR and 95% CI) between total maternal vitamin D intake and rapid growth from birth to 12 months and from birth to 2 years, using reported anthropometrics

| Maternal vitamin D intake | Weight growth from birth | | Height growth from birth | |
|---|---|---|---|---|
| | 12 months | 2 years | 12 months | 2 years |
| | OR (95% CI) | OR (95% CI) | OR (95% CI) | OR (95% CI) |
| **All mothers** | n=39 296, 59%* | n=18 360, 27% | n=31 640, 47% | n=14 946, 22% |
| <5 µg (ref) | | | | |
| 5–9.9 µg | 0.91 (0.85 to 0.97) | 1.03 (0.93 to 1.15) | 1.05 (0.97 to 1.13) | 1.01 (0.90 to 1.14) |
| 10–15 µg | 0.93 (0.86 to 1.02) | 1.02 (0.90 to 1.16) | 1.02 (0.93 to 1.12) | 0.95 (0.83 to 1.09) |
| >15 µg | 0.90 (0.83 to 0.98) | 0.97 (0.85 to 1.10) | 1.00 (0.91 to 1.10) | 1.04 (0.90 to 1.19) |
| **Mothers with normal weight†** | n=25 982, 62% | n=12 043, 29% | n=20 658, 49% | n=9700, 23% |
| <5 µg (ref) | | | | |
| 5–9.9 µg | 0.88 (0.81 to 0.96) | 1.04 (0.91 to 1.18) | 1.01 (0.91 to 1.11) | 1.05 (0.91 to 1.21) |
| 10–15 µg | 0.85 (0.76 to 0.95) | 1.03 (0.89 to 1.21) | 0.94 (0.83 to 1.06) | 0.98 (0.82 to 1.17) |
| >15 µg | 0.86 (0.77 to 0.95) | 0.97 (0.84 to 1.13) | 0.94 (0.84 to 1.06) | 1.01 (0.85 to 1.20) |
| **Mothers with overweight‡** | n=12 240, 61% | N=5838, 29% | N=10 118, 15% | N=4851, 24% |
| <5 µg (ref) | | | | |
| 5–9.9 µg | 0.96 (0.86 to 1.08) | 1.08 (0.90 to 1.29) | 1.12 (0.99 to 1.28) | 1.00 (0.82 to 1.21) |
| 10–15 µg | 1.10 (0.95 to 1.27) | 1.01 (0.81 to 1.27) | 1.14 (0.97 to 1.34) | 0.87 (0.68 to 1.12) |
| >15 µg | 1.01 (0.87 to 1.18) | 0.91 (0.72 to 1.16) | 1.08 (0.91 to 1.28) | 1.08 (0.84 to 1.38) |

Effect estimates derived by logistic regression, adjusted for maternal education, parity, maternal milk and yoghurt intake, maternal fibre intake and child's gender. The model with all mothers was also adjusted for maternal prepregnancy BMI.
*Proportions of the eligible pregnancies with the reported anthropometrics available from birth to the given age.
†Mothers with prepregnancy BMI 18.5–24.9 kg/m$^2$.
‡Mothers with prepregnancy BMI ≥25 kg/m$^2$.
BMI, body mass index.

and adjusted, compared with the reference group (table 5). At 5 years, children of mothers with vitamin D intakes of 10–15 µg/day during pregnancy, had 0.91 (95% CI 0.84 to 0.99) lower adjusted odds of being overweight, compared with the reference group. At 8 years, there was no association found for any intake group. For children of mothers with overweight, we found an increased risk of overweight at 5 years (OR 1.09, 95% CI 1.01 to 1.18) and at 8 years (OR 1.12, 95% CI 1.02 to 1.23) for children of mothers with vitamin D intakes of 5–9.9 µg/day, compared with the reference group (table 5). However, no association between maternal vitamin D intake and risk of overweight was found for children of mothers with intakes of 10–15 or >15 µg/day at 3 years, 5 years or 8 years of age.

In the sensitivity analyses on childhood overweight using the reported weights and heights, the results for the children of mothers with normal weight, with vitamin D intake >15 µg/day persisted (OR=0.87, 95% CI 0.77 to 0.98) (online supplemental table 10). Also, for children of mothers with overweight, with total vitamin D intake of 5–9.9 µg/day, a significant odds ratio (OR 1.22, 95% CI 1.06 to 1.40) was obtained. After exclusion of the highest 90th percentile of maternal vitamin D intakes, similar results as in the main models were obtained, both in children of mothers with normal weight and with overweight. Adjusting for birth weight did not change the direction of the associations, neither in children of mothers with normal weight, nor in children of mothers with overweight.

### Siblings with discordant total maternal vitamin D intake

When comparing weight and height growth trajectories of discordant siblings, the analyses showed similar estimates as the main models, but non-significant differences (online supplemental tables 5 and 6). The growth trajectories did not differ between siblings with discordant maternal vitamin D intake. When comparing the risk of childhood overweight at 3 years, 5 years and 8 years in discordant siblings, the same effect estimates as for the main models were obtained, however, not significant (online supplemental table 7).

### DISCUSSION

In this longitudinal study of the associations between maternal vitamin D intake and child growth, we identified

**Table 5** Associations (OR and 95% CI) between total maternal vitamin D intake and risk of childhood overweight at 3, 5 and 8 years, using predicted anthropometrics

| Maternal vitamin D intake | 3 years OR (95% CI) | 5 years OR (95% CI) | 8 years OR (95% CI) |
|---|---|---|---|
| **All mothers n=66 840** | | | |
| <5 µg (ref) | | | |
| 5–9.9 µg | 0.99 (0.93 to 1.06) | 1.03 (0.98 to 1.08) | 1.10 (1.03 to 1.17) |
| 10–15 µg | 0.93 (0.86 to 1.01) | 0.97 (0.91 to 1.04) | 1.02 (0.94 to 1.11) |
| >15 µg | 0.97 (0.89 to 1.06) | 1.00 (0.94 to 1.06) | 1.09 (1.01 to 1.19) |
| P value for trend* | 0.222 | 0.553 | 0.098 |
| **Mothers with normal weight† n=41 970** | | | |
| <5 µg (ref) | | | |
| 5–9.9 µg | 0.92 (0.84 to 1.01) | 0.97 (0.91 to 1.04) | 1.07 (0.98 to 1.17) |
| 10–15 µg | 0.86 (0.77 to 0.97) | 0.91 (0.84 to 0.99) | 0.98 (0.87 to 1.09) |
| >15 µg | 0.88 (0.79 to 0.99) | 0.94 (0.87 to 1.02) | 1.05 (0.94 to 1.17) |
| P value for trend* | 0.015 | 0.039 | 0.920 |
| **Mothers with overweight‡ n=20 080** | | | |
| <5 µg (ref) | | | |
| 5–9.9 µg | 1.07 (0.97 to 1.18) | 1.09 (1.01 to 1.18) | 1.12 (1.02 to 1.23) |
| 10–15 µg | 1.01 (0.89 to 1.14) | 1.05 (0.95 to 1.16) | 1.06 (0.94 to 1.20) |
| >15 µg | 1.06 (0.93 to 1.20) | 1.03 (0.93 to 1.14) | 1.11 (0.98 to 1.26) |
| P value for trend* | 0.513 | 0.443 | 0.156 |

Effect estimates derived by logistic regression, adjusted for maternal education, parity, maternal milk and yoghurt intake, maternal fibre intake and child's gender. The model with all mothers was also adjusted for maternal prepregnancy BMI.
*P value for linear trend, obtained by regression with maternal vitamin D intake in categories as continuous variable.
†Mothers with prepregnancy BMI 18.5–24.9 kg/m$^2$.
‡Mothers with prepregnancy BMI ≥25 kg/m$^2$.
BMI, body mass index.

prepregnancy BMI as a major modifying factor. Among mothers with normal prepregnancy BMI, a maternal vitamin D intake ≥10 µg/day was associated with lower weight and height growth trajectories starting in infancy and persisting in toddlerhood. Higher maternal vitamin D intake was also associated with lower odds of a rapid weight gain during the first year of life and of child overweight in preschool years. The results indicated associations in opposing directions in children of mothers with prepregnancy overweight. The discordant sibling analysis revealed similar but weaker associations, possibly due to small sample size and supports the hypothesis. Our findings suggest that in children of normal weight mothers, vitamin D intake during pregnancy has a protective effect against rapid growth and overweight, perhaps due to a more favourable weight growth trajectory in infancy.

The main strengths of this study are the longitudinal design and the large, nationwide[33] study sample. The FFQ, used to capture the dietary vitamin D intake, was customised for assessing diet in the first half of pregnancy. Validation studies on the FFQ concluded the questionnaire to provide adequate estimates of vitamin D intakes from foods and supplements.[39 40] Another strength was the use of a growth curve model,

which is an approach to compensate for the lost to follow-up, by predicting weights and heights for all children. The similar estimates found in the sensitivity analyses of the reported weights and heights of the children bring some validity in the predicted anthropometrics of growth trajectories and childhood overweight. In addition, the drawing of a directed acyclic graph reduced the risk of confounding bias, since the models were adjusted for relevant covariates. Our study also has several weaknesses. First, maternal anthropometrics, child anthropometrics and food intake measures are self-reported and prone to misreport.[39 52 53] Although the women were not aware of the outcome of interest, maternal weight status and other characteristic are likely to influence the accuracy of self-reported information. It is known that overweight and obese women misreport their bodyweight and food intake to a larger degree than normal weight women, pertaining also to pregnant women.[52] Although self-reported weight and height encompass inaccuracy, self-report is a cost-effective alternative in large cohorts and the inaccuracy is not likely to largely bias the results.[54] It is also possible that the association we found could be explained by another

food, nutrient or behaviour related to the maternal vitamin D intake that was not taken into account, or residual confounding.

The protective effect of maternal vitamin D in pregnancy on the risk of overweight in children from mothers with prepregnancy normal weight is consistent with previous findings from a Dutch[23] and a Spanish study.[26] However, previous studies investigating the association between maternal vitamin D status and childhood growth or overweight, have explored the association by adjusting for prepregnancy BMI. To simply adjust for prepregnancy BMI could potentially fail to capture the variation in maternal body size and thereby the magnitude of the distribution and dilution of vitamin D in the body.[5] This may explain the somewhat opposing results found in previous studies on the association on maternal vitamin D status and child's growth, anthropometrics and risk of overweight.[23 24 26–30]

Several possible pathways behind the association between maternal vitamin D intake, growth and risk of overweight in childhood have been suggested. Some studies have for example found associations between vitamin D deficiency and increased maternal insulin resistance in pregnancy and gestational diabetes mellitus.[55–57] It is suggested that if the fetus is exposed to hyperglycaemic in utero, the fetal insulin production may be altered and the risk of type 2 diabetes later in life increased.[58] Also, children born by mothers affected by gestational diabetes mellitus are more likely to be overweight.[59] If low vitamin D intake in pregnancy increases maternal insulin resistance, this may be one mechanism programming the fetus towards increased susceptibility of metabolic diseases including overweight in childhood.

Our results underscore the importance of adequate vitamin D intake for the pregnant woman on growth in childhood. Moreover, our results highlight the significant contribution of supplements to reach the recommended vitamin D intake in a population with relatively low intake from foods. These findings suggest that maternal vitamin D intake is a potential modifiable factor for prevention of rapid postnatal growth in children of normal weight mothers. Since there is a lack of previous studies investigating the association between maternal vitamin D intake and childhood growth and overweight, further studies will need to replicate our results before the relevance can be fully determined.

## CONCLUSION
Our findings supports that vitamin D intake during pregnancy affects offspring postnatal growth, and that a higher vitamin D intake during the pregnancy has a protective effect against the risk of childhood overweight but only in children of mothers with normal prepregnancy weight. In pregnancy, achieving the recommended vitamin D intake may reduce the risks of overweight in children of mothers with normal weight.

**Acknowledgements** The Norwegian Mother, Father and Child Cohort Study is supported by the Norwegian Ministry of Health and Care Services and the Ministry of Education and Research. The authors thank the MoBa cohort funders and all the participating families in Norway for their valuable contribution.

**Contributors** HA initiated the study, all authors contributed with planning the study, ALB is responsible for data protection and access, AA conducted the statistical analyses and wrote the first version of the manuscript, EP assisted with the statistical analyses. All authors (HA, AA, EP, ALB, LL, AW and HMM) were involved in interpretation of the results and writing of the final manuscript.

**Funding** The current study was funded by the Swedish Research Council for Health, Working Life and Welfare (grant number 2018-00441) and the Sahlgrenska Academy (U2018/162). EP was funded by the Research Council of Norway, under the MILJØFORSK programme (project number 268465).

**Competing interests** None declared.

**Patient consent for publication** Consent obtained directly from patient(s)

**Ethics approval** The Regional Committees for Medical and Health Research Ethics. 2019/770 REK Nord.

**Provenance and peer review** Not commissioned; externally peer reviewed.

**Data availability statement** Data may be obtained from a third party and are not publicly available. No additional data are available. All data from the MoBa study are available to all qualified researchers/research groups in Norway and to international researchers who are collaborating with a Norwegian researcher.

**ORCID iDs**
Anna Amberntsson http://orcid.org/0000-0003-3510-9789
Anne Lise Brantsaeter http://orcid.org/0000-0001-6315-7134

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
