## [Reviewer comments · BMJ Open]

ARTICLE DETAILS

TITLE (PROVISIONAL)	Maternal vitamin D intake and BMI during pregnancy in relation to child's growth and weight status from birth to 8 years - a large national cohort study
AUTHORS	Amberntsson, Anna; Papadopoulou, Eleni; Winkvist, Anna; Lissner, Lauren; Meltzer, Helle Margrete; Brantsaeter, Anne Lise; Augustin, Hanna

VERSION 1 – REVIEW

REVIEWER	Hocher, Berthold IFLB, Nephrology
REVIEW RETURNED	06-May-2021

GENERAL COMMENTS	Well done study. However, some points need to be addressed: 1. Vitamin D intake is difficult to measure. It does not simply depend on consumption of vitamin D supplements/pills, it also depends on eating characteristics (fish consumption for example) and sun shine exposure. Did lower vitamin intake as estimated in your study affect blood vitamin D concentrations? It would be important to show this at least in a subset of your study to validate the method used in your study to estimate vitamin D intake.2. Vitamin D plasma levels in pregnant women depend on the socio-economic status of the mothers, see Reichetzeder et al. Kidney Blood Press Res. 2014;39(4):315-29. doi: 10.1159/000355809. Thus the socioeconomic status (maternal education level, family income) needs also be considered in all analysis you did.3. Maternal Vitamin D deficiency is linked to low birth weight, preterm birth, lower APGAR score, see Reichetzeder et al. Kidney Blood Press Res. 2014;39(4):315-29. doi: 10.1159/000355809. . Discuss this study in comparison to your data. Was birth weight, gestational age and APGAR score also affected in your study population?4. Vitamin D deficiency is linked to insulin resistance particularly in women (Chen et al., J Clin Endocrinol Metab. 2021 Apr 1:dgab213. doi: 10.1210/clinem/dgab213.). Maternal insulin resistance is thought to be one of the key mechanisms of fetal programming of offspring's phenotype (growth, obesity and also insulin resistance), see Tian et al., J Hypertens. 2019 Nov;37(11):2123-2134. doi: 10.1097/HJH.0000000000002156. Please discuss this in view of your results by referring to the mentioned papers.
--

	5. You should describe molecular actions of vitamin D during pregnancy on the growing fetus by referring to J Steroid Biochem Mol Biol. 20
--	--

REVIEWER	Reesukumal, Kanit Mahidol University
REVIEW RETURNED	14-Jun-2021

GENERAL COMMENTS	1. What is the reason for dividing study population into four categorized groups depend on estimated vitamin D intake? 2. What is the reason for doubling the estimated vitamin D value in group with vitamin D intake from supplements compare to group with vitamin D intake from foods alone? 3. Sun exposure can be of some value in determining vitamin D status and it is nice to know how many hour per day this occurred especially in summer. The authors may want to note this and reference this as well.
---

VERSION 1 – AUTHOR RESPONSE

Reviewer 1 - Dr. Berthold Hoher, IFLB. Comments to the Author:

Comment 1. Vitamin D intake is difficult to measure. It does not simply depend on consumption of vitamin D supplements/pills, it depends also on eating characteristics (fish consumption for example) and sun shine exposure. Did lower vitamin intake as estimated in your study affect blood vitamin D concentrations? It would be important to show this at least in a subset of your study to validate the method used in your study to estimate vitamin D intake.

Response: Thank you so much for all your comments. We have now elaborated on the validity of the method used for estimating the vitamin D intake in the methods section by providing more details about the previously published validation study (page 7). There, we explain that supplement use was the dominant predictor of plasma 25OHD and that also season played a role.

Comment 2. Vitamin D plasma levels in pregnant women depend on the socio-economic status of the mothers, see Reichetzeder et al. Kidney Blood Press Res. 2014;39(4):315-29. doi: 10.1159/000355809. Thus the socioeconomic status (maternal education level, family income) needs also be considered in all analysis you did.

Response: You raise an important issue here and we have included maternal education as a proxy for socio-economic status in all analyses performed. We have made this clearer in the statistical section (page 9). Since education can be regarded as the determinant of income, we included only maternal education (and not income) to avoid collinearity within the models.

Comment 3. Maternal Vitamin D deficiency is linked to low birth weight, preterm birth, lower APGAR score, see Reichetzeder et al. Kidney Blood Press Res. 2014;39(4):315-29. doi: 10.1159/000355809. Discuss this study in comparison to your data. Was birth weight, gestational age and APGAR score also affected in your study population?

Response: As presented in Table 1 in our manuscript, birth weight did differ between the categories of maternal vitamin D intake. The women with a vitamin D intake <5µg/day gave birth to children with

a 10g higher median birth weight compared to the other categories. All other categories shared the same median birth weight. Gestational age did not differ between the categories in our study, also shown in table 1. Unfortunately, we were not able to not investigate potential differences in APGAR-score between categories of maternal vitamin D intake, since we did not have access to information about the APGAR-scores. We have now clarified in the methods section (page 9) that birth weight in fact was investigated as a confounder in the mixed effect models. Birth weight was not included in the final model as it did not affect the estimates. Birth weight was also investigated as an interaction factor, but did not modify the association. In our study population, 1356 (2%) children were born with a low birth weight (<2500g), of which 451 (33%) were children of mothers with vitamin D intake <5µg/day. Also, 2594 (3.9%) were born preterm (before 37 completed gestational weeks), of which 840 (23%) were children of mothers with vitamin D intake <5µg/day. Thus, it is unlikely that these groups are driving the observed associations in the whole MoBa population. Additionally, the rapid growth measure take birth weight into account as it is a function of it. Since the main aim of the study was not to investigate outcomes at birth, we hope that it is sufficient only to clarify that birth differed significantly between the lowest category of vitamin d intake and the other intake categories, but was not further discussed in the manuscript

It is interesting though to see that Reichetzeder et al. report that low 25OHD levels are associated to low birth weight after controlling for gestational age, and that low maternal 25OHD levels also impacted the APGAR score.

Comment 4. Vitamin D deficiency is linked to insulin resistance particularly in women (Chen et. al, . J Clin Endocrinol Metab. 2021 Apr 1:dgab213. doi: 10.1210/clinem/dgab213.). Maternal insulin resistance is thought to be one of the key mechnisms of fetal programming of offspring`s phenotype (growth, obesity and also insulin resistance), see Tian et al., J Hypertens. 2019 Nov;37(11):2123-2134. doi: 10.1097/HJH.0000000000002156. Please discuss this in viw of your results by referring to the mentioned papers.

Response: A brief discussion of this potential pathway has been added to the discussion (page 24).

Comment 5. You shoud describe molecular actions of vitamin D during pregnancy on the growing fetus by referring to J Steroid Biochem Mol Biol. 2018 Jun;180:51-64. doi: 10.1016/j.jsbmb.2017.11.008.

Response: A brief paragraph on the molecular actions of vitamin D in pregnancy has been added in the introduction (age 5).

Reviewer 2 - Dr. Kanit Reesukumal, Mahidol University Comments to the Author:

Comment 1. What is the reason for dividing study population into four categorized groups depend on estimated vitamin D intake?

Response: Thank you so much for your important comments. As shown in the validation study (referred to at page 7) of the food frequency questionnaire (FFQ) used to estimate vitamin D intake, the FFQ had a fair ability to rank individuals by nutrients. An FFQ is a rather crude instrument that is better suited for ranking individuals than for precise estimation. The usage of categorized vitamin D therefore has a higher precision than the continuous vitamin D variable.

Comment 2. What is the reason for doubling the estimated vitamin D value in group with vitamin D intake from supplements compare to group with vitamin D intake from foods alone?

Response: We would have preferred to use the recommended daily intake of vitamin D according to the Nordic Nutrition Recommendations. However, these cut offs were not possible to use, when

investigating foods alone, since the vitamin D intakes from foods generally were much lower. Thus, the number of individuals in the higher intake categories became too few. Therefore, study specific cut-offs were used when studying vitamin D intake from foods alone, as compared to total vitamin D intake.

Comment 3. Sun exposure can be of some value in determining vitamin D status and it is nice to know how many hour per day this occurred especially in summer. The authors may want to note this and reference this as well.

Response: We have now elaborated on season in the paragraph describing the validation study by providing more details in the method section (page 7). In the validation study, vitamin D supplement use was found to be the dominant predictor of plasma 25OHD and was therefore suggested to sufficiently reflect the 25OHD concentration. It is not possible to estimate sun exposure within the MoBa study.

VERSION 2 – REVIEW

REVIEWER	Hocher, Berthold IFLB, Nephrology
REVIEW RETURNED	30-Aug-2021

GENERAL COMMENTS	my points are adressed, no further comments
---

REVIEWER	Reesukumal, Kanit Mahidol University
REVIEW RETURNED	16-Sep-2021

GENERAL COMMENTS	This large and nation wide cohort study by questionnaire survey have been shown the association of dietary vitamin D intake during the first half of pregnancy and child growth with pre-pregnancy BMI as a major modifying factor. However, further studies with minimizing of confounding will need to validate the relevant results.
---